# Applying the Moving Epidemic Method to Establish the Influenza Epidemic Thresholds and Intensity Levels for Age-Specific Groups in Hubei Province, China

**DOI:** 10.3390/ijerph19031677

**Published:** 2022-02-01

**Authors:** Yuan Jiang, Ye-qing Tong, Bin Fang, Wen-kang Zhang, Xue-jie Yu

**Affiliations:** 1State Key Laboratory of Virology, School of Public Health, Wuhan University, Wuhan 430071, China; jiangy971004@163.com (Y.J.); 13843205591@163.com (W.-k.Z.); 2Hubei Provincial Center for Disease Control and Prevention, Wuhan 430079, China; t_yeqing@163.com (Y.-q.T.); nicolfang@163.com (B.F.)

**Keywords:** moving epidemic method, school-aged children, influenza surveillance

## Abstract

Background: School-aged children were reported to act as the main transmitter during influenza epidemic seasons. It is vital to set up an early detection method to help with the vaccination program in such a high-risk population. However, most relative studies only focused on the general population. Our study aims to describe the influenza epidemiology characteristics in Hubei Province and to introduce the moving epidemic method to establish the epidemic thresholds for age-specific groups. Methods: We divided the whole population into pre-school, school-aged and adult groups. The virology data from 2010/2011 to 2017/2018 were applied to the moving epidemic method to establish the epidemic thresholds for the general population and age-specific groups for the detection of influenza in 2018/2019. The performances of the model were compared by the cross-validation process. Results: The epidemic threshold for school-aged children in the 2018/2019 season was 15.42%. The epidemic thresholds for influenza A virus subtypes H1N1 and H3N2 and influenza B were determined as 5.68%, 6.12% and 10.48%, respectively. The median start weeks of the school-aged children were similar to the general population. The cross-validation process showed that the sensitivity of the model established with school-aged children was higher than those established with the other age groups in total influenza, H1N1 and influenza B, while it was only lower than the general population group in H3N2. Conclusions: This study proved the feasibility of applying the moving epidemic method in Hubei Province. Additional influenza surveillance and vaccination strategies should be well-organized for school-aged children to reduce the disease burden of influenza in China.

## 1. Introduction

Human influenza has become a severe global health problem in recent years. According to the World Health Organization estimation, influenza can cause 3–5 million severe illnesses and approximately 300,000 deaths each year [1]. The World Health Organization re-emphasizes the importance of influenza monitoring and has established a set of standards for sentinel surveillance after suffering great losses in both the economy and people’s health during the influenza A (H1N1) pandemic in 2009 [2]. Influenza surveillance networks for monitoring different indicators or targeting different populations are being set up worldwide [3,4]. The sanitation department collects and processes these data to provide policymakers with feasible prevention and control strategies.

The influenza vaccination program was considered a cost-effective measure and was recommended as the priority method against influenza. However, the accurate prediction of the start of the epidemic was correlated with the protective effect of the vaccine [5,6]. Setting up the epidemic threshold is one of the methods for disease prediction [7]. Several studies have provided empirical or statistical methods to establish the epidemic threshold, such as the Serfling regression model [8] and the moving logistic regression method [9], or defined the epidemic threshold as 10% of the influenza detection rate [10]. However, these methods cannot evaluate the intensity levels between seasons and have requirements for the form of data. In addition, the changes of epidemic patterns in different seasons led to the invalidity of a fixed proportion of the influenza detection rate. The moving epidemic method (MEM) was first put forward in Spain [11] and has been adopted as a routine surveillance program to establish the epidemic threshold and intensity levels in some temperate countries [12,13]. Increasing studies in other climate zones have also demonstrated the high goodness of fit of the threshold established by the MEM [14,15,16,17]. In China, there is no standard method for setting epidemic thresholds, so introducing a practical method for early influenza detection will be helpful to reduce the disease burden for Chinese people.

Children are considered to be one of the high-risk populations for influenza. A previous study demonstrated that the school-aged children had a higher incidence rate than adults [18], and school-aged children might have acted as the primary transmitter during the epidemic [19]. A contact mode study in eight European countries indicated that it might be due to the high contact frequency in this age group [20]. Offering the start time of the influenza epidemic in school-aged children can help the sanitation department take early and proper measures and reduce the disease burden in the whole population. However, most MEM studies have only concentrated on the general population. The epidemic threshold of school-aged children has not been discussed.

In this study, we analyzed the characteristics of influenza in Hubei Province from the season 2010/2011 to 2018/2019. Subsequently, we established the epidemic thresholds and intensity levels for the school-aged populations and compared the goodness of fit between the models established by different age group data. The feasibility of the moving epidemic method was evaluated by the cross-validation process.

## 2. Materials and Methods

### 2.1. Case Identification

Influenza-like illness (ILI) was described as a patient whose body temperature was above 38 °C with either cough or sore throat, with the lack of other experimental diagnoses. A period of high body temperature should happen in the course of the acute fever.

### 2.2. Sample Source

Hubei Province is a subtropical area located in Central China (108°21′–116°07′ E, 29°01′–33°6′ N), including twelve cities, one prefecture and four administrative cities directly under the jurisdiction of the province. It covers an area of 185,900 km^2^, with 55,750,000 permanent residents, including more than 8,610,000 school-aged children.

Sentinel hospitals were set up in 2000 to collect influenza-like illness visitors’ information, and the locations of the national sentinel hospitals are shown in Figure 1. The absence of a sentinel hospital in some cities was due to the small size of the population (Shen Nong Jia) or that they were directly under the governance of Hubei Province (Qianjiang City, Tianmen City and Xiantao City). Each sentinel hospital was asked to collect 20 swab samples (including throat swabs, nasal swabs and nasopharyngeal swabs) per surveillance week from ILI patients. The swab samples were collected in sampling tubes and stored at 2–8 °C and then transported to the laboratory of a local municipal center for disease control and prevention (CDC) for further testing. Real-time reverse transcription-polymerase chain reaction (RT-PCR) was used to identify the subtypes of the influenza virus, and the results were submitted to the national influenza surveillance network. The quality of the test results of each city was reviewed by the influenza reference laboratory of Hubei CDC.

### 2.3. Data Collection

In mainland China, children above three years old are able to attend kindergarten. According to the living environment and social contact mode, we roughly divided the whole population into three age groups. We defined the population below three years old as pre-school children and the population above eighteen years old as adults, while the rest were defined as school-aged children, who spent most of the time in a classroom and had more opportunities to interact with others.

In Hubei Province, all three influenza strains occurred in the winter, while influenza A(H3N2) intermittently occurred in the summer. To include the entire prevalence curve of each strain, we defined the epidemic surveillance period of Hubei Province as week 40 to week 41 of the following calendar year. This study obtained the surveillance data from season 2010/2011 to 2018/2019, with 468 epidemic weeks. Due to the technical limitations, the influenza A strains in the first several surveillance seasons contained many untyped samples. We did not further divide these data like in another study [21] to avoid potential confounding factors.

### 2.4. Moving Epidemic Method

The MEM procedure could be divided into three steps: The first step was to define the epidemic period. The maximum accumulated rates percentage method (MAP) was applied to divide the season into the pre-epidemic, epidemic, and post-epidemic periods. The epidemic period defined by this algorithm was the real epidemic period. The details can be found in a previous study [11]. Parameter δ plays a vital role in this step, as this is a pre-defined parameter recommended to range from 2.0% to 4.0%, and the proper value was chosen by the criteria of the highest Youden’s Index (YI). The second step was to establish the epidemic and the intensity thresholds. The epidemic thresholds were calculated as the upper limits of the 95% one-sided confidence interval of the arithmetic mean of the 30 highest weekly values of the pre-epidemic and post-epidemic periods. The epidemic intensity thresholds were calculated as the upper limits of the 40%, 90% and 97.5% one-sided confidence intervals of the geometric mean of the 30 highest weekly values of the epidemic period. The medium intensity, the high intensity and the very high intensity were defined as 40%, 90% and 97.5%, respectively. The final step was to assess the performance of the model. The MEM adopted a cross-validation procedure to estimate the epidemic thresholds and assess the goodness of fit. This cross-validation procedure was performed by extracting every epidemic season from the historical data and using the remaining seasons to calculate the threshold of the extracted one. When the weekly positive rate exceeded the epidemic threshold for two consecutive weeks, the first week was defined as the alert week. The timeliness was the number of weeks between the alert week and the first week of the epidemic period defined in the first step. If the alert week was after the real start week, it was defined as the detection lag. Sensitivity (Se) was calculated as the number of weeks above the epidemic and post-epidemic thresholds divided by the number of real epidemic weeks, and specificity (Sp) was defined as the number of weeks below the epidemic and post-epidemic thresholds divided by the number of real nonepidemic weeks. Similar indicators, such as the positive predictive value (PPV), negative predictive value (NPV) and Youden’s Index (YI = Sp + Se − 1), were further developed to assess the performance of the model.

Influenza epidemics occur twice per season in Hubei Province. We applied the two-wave transformation tool offered by the R package “memapp” [22] to separate these double-peak data. However, a three-peak epidemic pattern was observed in season 2016/2017. We excluded this season from calculating the total influenza thresholds, because the transformation tool could not separate this season. For the type-specific strain, we followed the examples of the study in Guangdong Province and Scotland [17,23], and the epidemic threshold was calculated using weekly data in the seasons when the proportion of the strain exceeded 20%. We defined the last season as the test set for each influenza strain, while the other was the training data. The MEM package version 2.16 of R language software performed the procedure.

### 2.5. Statistical Method

The influenza positive rates and the proportions of the different age groups were compared using the Pearson chi-square test. The intensity levels, the average start week and the duration were compared by the Mann–Whitney *U* test. These procedures were conducted by IBM SPSS Statistics 25 (SPSS Inc., Chicago, IL, USA).

## 3. Results

### 3.1. Description of Influenza Activity in Hubei Province

From season 2010/2011 to 2018/2019, the sentinel hospitals identified 927,585 ILI patients, and 139,745 of them were tested by the laboratory. The Pearson correlation test provided a poor correlation between the weekly ILI proportion and positive rate, with a coefficient of 0.39 (*p* < 0.001). The pre-school, school-aged children and adults respectively accounted for 30.5%, 40.7% and 28.8% of all the samples sent for testing. Among all the laboratory-tested samples, 18,666 samples were positive (13.4%), and the positive rates of the three age groups were 8.8%, 17.4% and 12.5%, respectively. Gender did not affect the opportunity to become infected (chi-square test, *p* > 0.05). Suizhou City had the highest positive proportion, and the chi-square test for pairwise comparison showed that the positive rates of Suizhou, Ezhou, Huangshi, Yichang, Jingmen and Huanggang City were significantly higher than those of the other cities in Hubei Province (Table 1). The chi-square test also demonstrated that the positive rate increased when the children became older, starting from one year old and reaching the highest level at 6–12 years old, while no significant difference was found between age groups above 18 years old (Figure 2). H3N2 was the main-type influenza in pre-school children (45.0%) and adults (42.1%), this proportion was significantly higher in pre-school children (*p* = 0.008). Most of the school-aged children were infected by influenza B (42.5%), higher than the other two age groups (*p* < 0.001). The proportion of H1N1 had no significant difference among the three age groups (*p* = 0.60).

Fourteen epidemic peaks were observed during all the surveillance seasons. Four of the nine seasons presented more than one peak. Notably, most of the seasons with multiple peaks were due to cocirculated influenza (Figure 3). H1N1 and influenza B provided one peak per season, while seasonal H3N2 in Hubei Province had two peaks, and the activity in the winter was milder than in the summer. The epidemic onset time and the length of H1N1 and the summer peak of H3N2 between the different seasons were less varied than influenza B and the winter peak of H3N2.

### 3.2. Epidemic Thresholds and Intensity Levels Established by the MEM Model

Table 2 summarized the laboratory results for school-aged children in different seasons, and Table 3 presented the epidemic thresholds for school-aged children. According to the maximum Youden’s Index, the best parameters δ for total influenza, H1N1, H3N2 and influenza B were defined as 2.0, 3.1, 2.5 and 2.0, respectively. The epidemic threshold for total influenza in season 2018/2019 was 15.74%, and the alert week was defined as week 52, the same week as the start of the epidemic period defined by the MAP method, indicating that the epidemic detection timeliness was 0 (Figure 4). During the threshold establishment, the epidemic threshold of influenza B was significantly higher than influenza H1N1 and H3N2 (*p* < 0.001), while H1N1 and H3N2 had similar thresholds (*p* = 0.184). For the total strain, influenza typically started (median start week) at week 50 in the winter and week 26 in the summer. For the type-specific strain, typical H1N1 started at week 2, influenza B started at week 49 and H3N2 started at week 47 in the winter and 27 in the summer. Influenza B had the most prolonged epidemic duration (influenza B versus H1N1, *p* = 0.004; influenza B versus H3N2, *p* = 0.001), and the durations of H1N1 and H3N2 were similar (*p* = 0.599). We observed that the median duration of summer H3N2 was 2 weeks longer than that in the winter, but this difference was not significant (*p* = 0.072).

We also established the epidemic and intensity thresholds of the general population and other age-specific groups. The laboratory results are shown in Appendix A, while the intensity levels and other epidemic details are shown in Appendix B. For the general population, the alert week was one week earlier in season 2018/2019. In order to compare the differences in the timing of the onset of influenza among the different age groups, we selected and compared the common epidemic seasons of the three age groups. We observed that the epidemic descriptions of the same season between the different age groups were varied. In addition, by comparing the median start week, we found H1N1 and H3N2 struck first among adults, while influenza B started first among school-aged children. Pre-school children were the last to get infected and had a relatively shorter epidemic duration than the other two age groups in all three strains. However, the differences of the start week and the duration between the age groups were not statistically significant (*p* > 0.05).

### 3.3. MEM Goodness of Fit

In Table 4, we present the goodness of fit of the MEM model established for different age groups. All groups performed well, with high sensitivity and specificity. In most cases, school-aged children had the highest Youden’s Index. However, pre-school children and adults seemed to have worse performances than the general population.

For the type-specific strains, we observed that the performance of influenza B was relatively worse. By inspecting the goodness of fit of each season, we found that the inclusion of the season with low influenza activity led to a poor outcome. Therefore, we attempted to remove the 2010/2011 data from the training set of influenza B and compared the model performance before and after the exclusion (Appendix C). The results indicated that the start week and duration of influenza of the remaining seasons did not change after excluding the low influenza activity seasons, while the performance of influenza B was significantly improved.

## 4. Discussion

This study described the influenza epidemiology characteristics in Hubei Province from season 2010/2011 to 2018/2019, established the epidemic thresholds for the general and age-specific groups and assessed the MEM performance between these groups. Our study proved the feasibility of applying the MEM in Hubei Province and proposed that the school-aged children could replace the general population for influenza routine surveillance. To our knowledge, this study is one of the first attempts in applying the MEM to establish the thresholds for age-specific groups in the subtropical region.

Choosing proper indicators can help with improving the effectiveness of the model. This study applied the virology data rather than the ILI incidence rate to fit the MEM. Previous studies have proved the feasibility of a series of indicators to establish the influenza prediction model, including the ILI incidence rate or cases [24], virology data [21], excess mortality rate [25] and school absenteeism rate [26]. In Hubei Province, a poor correlation was found between the positive and ILI incidence rates, and the ILI seemed to provide less seasonality. ILI could be affected by multiple factors, such as the coinfection of other respiratory viruses with similar symptoms, the willingness of parents to send their children to medical care and the criteria used to define ILI. These findings suggested that the ILI incidence rate may not be a suitable indicator for establishing the influenza model in tropical and subtropical areas, which was in line with other studies. [27,28].

For the general population, the influenza epidemic threshold in season 2018/2019 was defined as 12.10%, lower than the thresholds established for all subtropical cities in China [29]. A difference was also found between the epidemic thresholds for Guangdong Province and its climate zone [17]. These results may be attributed to the large territory in China, which leads to climate and economic differences among provinces, finally resulting in various influenza epidemic patterns of regions even in the same climate zones. However, we found that, when there was a double-peak influenza pattern in Hubei Province, a similar phenomenon could also be observed in all studied cities of Hubei Province, implying that the epidemiological differences between cities may not be as great as the differences at the provincial level. This assumption was also supported by Yu and his colleagues [30], which made it possible to establish the influenza thresholds at the provincial level. Nevertheless, we also observed that the positive rate of influenza among cities in Hubei Province was varied, ranging from 9.18% to 19.06%. Nevertheless, using virology data to establish the epidemic thresholds for each city was not feasible due to the lack of ILI samples sent for laboratory testing per week. If we want to establish more accurate thresholds for influenza detection, we will have to increase the number of test samples, which requires a higher laboratory testing capacity. In the current situation, where the laboratory testing capacity was relatively fixed, adapting a specific population for testing may be one of the solutions. According to our observations, most test samples came from 1–6 year old, but the 6–12 age group had the highest infection rate. A similar phenomenon was also observed in Niger [31] and several states of the United States [24]. Moreover, the cross-validation analysis showed that school-aged children performed better in MEM. These findings suggest that school-aged children might be a worthy and suitable population to study the influenza prevalence. In Hubei Province, the comparison of the start week provided slight differences among the different age groups. This phenomenon was also observed in Canada and Korea [32,33], making it possible for countries that do not have enough laboratory testing capacity to substitute school-aged children for the whole population surveillance.

The concept of a school-based surveillance system was proposed 20 years ago [34], and its effectiveness has been proven in studies in different countries [26,35,36]. In China, school-based surveillance has been incorporated into community influenza surveillance. However, the current school-based surveillance system only concentrates on syndromic indicators such as absenteeism and ILI rates. Compared with other age groups, the student samples were easier to obtain and manage. Establishing a school-based virology surveillance system could help to quickly identify the circulating strain and utilize relatively small samples for influenza monitoring.

A vaccination program among school-aged children was recommended in the United States [18,37] and Europe [38]. In China, the health insurance system does not cover the influenza vaccination [39], partly responsible for the low levels of vaccine coverage. Our study revealed that influenza B was most prevalent in school-aged children. Quadrivalent influenza vaccination among this age group might have considerable efficacy. However, the timings of influenza B epidemics are highly variable, so accurate predictions of the onset of an epidemic is crucial for vaccination.

The MEM provided a simple approach for establishing the epidemic threshold to detect the epidemic start week and has a relatively low data requirement. A study in England suggested that the MEM was more suitable than the empirical percentile methods for monitoring the onset of an epidemic [40] and has been utilized for influenza routine surveillance in Europe. However, we found that seasons with low intensities could affect the MEM performance. Taking influenza B as an example, after excluding season 2010/2011 from the training set, all the indexes reflecting a goodness of fit increased significantly. These results suggested that the historical data should be filtered before applying it to the MEM model. The robustness of the MEM will be increased while using relatively stable historical data. However, no study has provided a formal method for season selection. Further research on the inclusion criteria should be conducted in the future.

This study had several limitations. First, the virology data of Hubei Province did not divide the B lineages into influenza B (Yamagata) and influenza B (Victoria), which have different timings of the onset and epidemic periods [41] and may lead to the false identification of the long and low intensities of influenza B activity. With the reduction in the cost of laboratory tests [42], more specific data was enabled to be well-documented. Second, establishing the epidemic thresholds by the MEM requires the training set to include enough seasons. However, while we applied the virology data of the type-specific strain, the seasons included in the training set of each strain were less than five seasons [11], which may have decreased the robustness of the model.

## 5. Conclusions

This study proved the feasibility of applying the MEM for influenza routine surveillance in Hubei Province, a subtropical region with a complex epidemic pattern. Compared with other age groups, the MEM model established with school-aged children’s data was more accurate in detecting the onset of the influenza epidemic. The importance of establishing a school-based influenza surveillance system should be noted by the sanitation department and the government.

## Figures and Tables

**Figure 1 ijerph-19-01677-f001:**
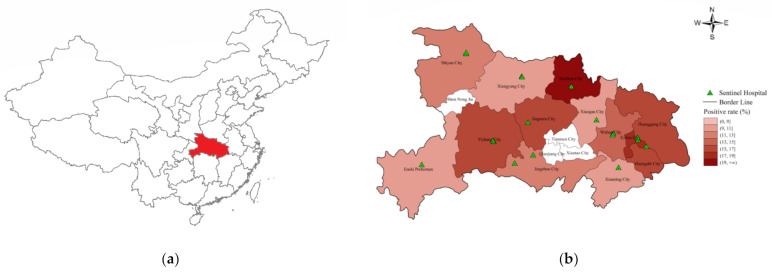
(**a**) The geographic location of Hubei Province in China. (**b**) The locations of influenza sentinel hospitals and positive rates of influenza in the sentinel hospitals in Hubei Province.

**Figure 2 ijerph-19-01677-f002:**
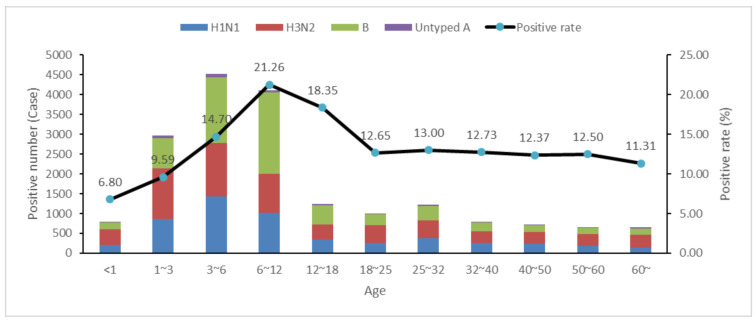
The case number and positive rate of the influenza subtypes in different age groups in Hubei Province, China (*n* = 18,666 cases). H1N1 and H3N2: influenza A (H1N1) and influenza A (H3N2) and B: influenza B.

**Figure 3 ijerph-19-01677-f003:**
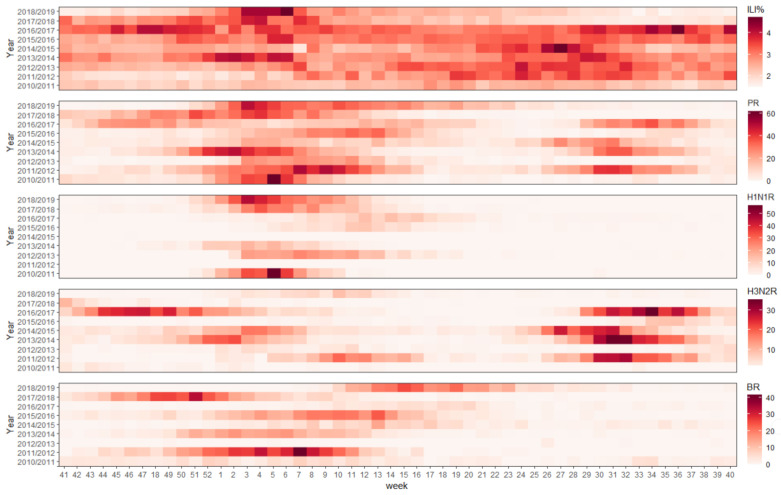
Influenza surveillance results from season 2010/2011 to 2018/2019. ILI%: weekly influenza-like illness patients’ proportion among all outpatient visitors. PR: weekly laboratory testing positive samples among all samples delivered by sentinel hospitals. H1N1R, H3N2R and BR: weekly laboratory testing positive rate for H1N1, H3N2 and influenza B. The color represented the value of the surveillance week.

**Figure 4 ijerph-19-01677-f004:**
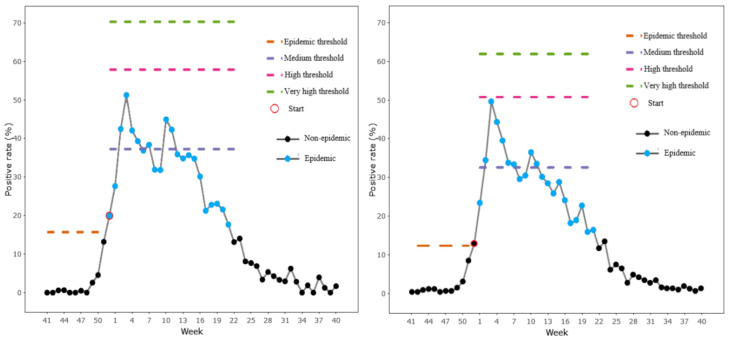
The positive rate, epidemic and intensity levels of school-aged children (**left**) and general population (**right**) in season 2018/2019 in Hubei Province, China.

**Table 1 ijerph-19-01677-t001:** Gender, subtypes and positive rate of influenza in Hubei Province, China from 2010/2011 to 2018/2019.

Characteristic	Specimen No.	Positive No.	Positive Rate (%)	H1N1 No.	H3N2 No.	B No.	Untyped A No.
Gender							
Male	75,738	10,231	13.51	2948	3548	3502	233
Female	64,007	8435	13.18	2364	2983	2909	179
Season							
2010/2011	8762	761	8.69	419	42	256	44
2011/2012	8995	2034	22.61	0	727	941	366
2012/2013	11,155	785	7.04	621	152	11	1
2013/2014	16,808	2815	16.75	526	1466	823	0
2014/2015	17,056	1834	10.75	1	1316	517	0
2015/2016	17,921	1804	10.07	409	208	1187	0
2016/2017	19,805	3106	15.68	601	2190	315	0
2017/2018	19,318	2573	13.32	1132	127	1314	0
2018/2019	19,925	2954	14.83	1603	303	1047	1
City							
Suizhou	8275	1577	19.06	372	471	733	1
Ezhou	8269	1432	17.32	540	325	554	13
Huangshi	8250	1326	16.07	248	582	487	9
Yichang	10,723	1677	15.64	496	511	603	67
Jingmen	7783	1203	15.46	296	455	438	14
Huanggang	8094	1251	15.46	337	504	407	3
Wuhan	20,698	2704	13.06	735	1188	596	185
Shiyan	17,408	2168	12.45	647	774	717	30
Jingzhou	16,975	2067	12.18	499	785	779	4
Xianning	8121	880	10.84	296	284	300	0
Xiaogan	7068	719	10.17	296	213	188	22
Xiangyang	10,554	971	9.20	319	250	353	49
Enshi	7527	691	9.18	231	189	256	15
Total	139,745	18,666	13.36	5312	6531	6411	412

**Table 2 ijerph-19-01677-t002:** Laboratory results of influenza in school-aged children in Hubei Province, China.

Season	Test ^1^	Positive	Identified ^2^	H1N1 ^4^		H3N2		Influenza B	
				Case	Prop (%) ^3^	Case	Prop (%)	Case	Prop (%)
2010/2011	3671	340	328	157	47.9	22	6.7	149	45.4
2011/2012	3845	1108	950	0	0.0	300	31.6	650	68.4
2012/2013	4175	407	407	325	79.9	81	19.9	3	0.7
2013/2014	6316	1297	1297	204	15.7	575	44.3	515	39.7
2014/2015	6319	872	872	0	0.0	569	65.3	303	34.7
2015/2016	6924	1070	1070	208	19.4	80	7.5	783	73.2
2016/2017	7661	1389	1389	347	25.0	850	61.2	196	14.1
2017/2018	8236	1600	1600	685	42.8	55	3.4	854	53.4
2018/2019	9592	1785	1785	864	48.4	186	10.4	737	41.3

^1^ Test: the number of samples tested by the laboratory. ^2^ Identified: the number of positive samples after excluding untyped influenza A. ^3^ Prop: proportion. ^4^ H1N1 and H3N2: influenza A (H1N1) and influenza A (H3N2).

**Table 3 ijerph-19-01677-t003:** Start week, duration and the intensity levels of influenza A and B among school-aged children in Hubei Province, China.

Subtype	Season	Start ^1^	Duration	E ^2^	M	H	VH	T	Description
H1N1 ^5^	2010/2011	1	9	6.11	23.73	39.55	49.56	0	Very high
	2012/2013	3	13	6.05	22.51	42.49	56.27	0	Medium
	2016/2017	8	11	5.71	26.04	42.74	53.21	0	Low
	2017/2018	2	12	3.74	22.02	40.25	52.54	−3	Medium
	2018/2019	51	13	6.05	21.56	38.11	49.03	0	High
H3N2 ^3^	2011/2012(W)	5	12	6.61	24.29	32.80	37.46	0	Medium
	2011/2012(S)	26	13	6.27	22.89	33.28	39.26	0	Medium
	2013/2014(W)	46	14	5.98	23.51	33.71	39.53	1	Medium
	2013/2014(S)	28	11	6.41	22.50	32.39	38.05	0	Medium
	2014/2015(W)	48	12	6.07	23.59	33.20	38.61	1	Medium
	2014/2015(S)	25	11	6.46	22.74	32.77	38.52	0	Medium
	2016/2017(W)	42	14	5.92	22.60	32.73	38.55	−1	Medium
	2016/2017(S)	29	11	6.38	22.39	31.37	36.42	0	Very high
B	2010/2011	41	26	10.11	25.79	43.71	55.18	NA ^4^	Low
	2011/2012	46	19	9.39	19.67	38.98	52.73	0	High
	2013/2014	49	16	10.84	21.40	47.69	67.96	1	Medium
	2014/2015	5	15	10.88	23.09	48.50	67.33	5	Low
	2015/2016	1	17	8.31	20.35	44.12	62.10	−5	Medium
	2017/2018	42	16	10.10	20.45	44.53	62.80	0	Medium
	2018/2019	10	15	10.66	20.86	46.17	65.60	1	Medium
Total	2010/2011	52	12	15.93	37.49	56.95	68.51	0	High
	2011/2012(W)	49	19	14.04	36.73	55.28	66.23	−3	High
	2011/2012(S)	26	13	16.01	37.59	58.35	70.86	3	Medium
	2012/2013	2	16	16.04	38.61	59.57	72.17	1	Low
	2013/2014(W)	48	17	15.77	36.89	56.32	67.90	1	Medium
	2013/2014(S)	28	12	15.98	39.20	59.30	71.20	2	Low
	2014/2015(W)	52	20	15.83	39.46	58.95	70.39	2	Low
	2014/2015(S)	24	14	15.65	39.32	58.85	70.32	2	Low
	2015/2016	1	17	15.89	37.24	57.68	69.99	0	Medium
	2017/2018	42	23	15.03	37.26	57.72	70.04	−1	Medium
	2018/2019	52	22	15.74	37.33	57.93	70.36	0	Medium

^1^ Start: the start week defined by the MAP method. ^2^ E: epidemic threshold. M: medium intensity. H: high intensity. VH: very high intensity. T: timeliness ^3^ S: the summer wave of H3N2. W: the winter wave of H3N2. ^4^ NA: the positive rate does not exceed the threshold for two consecutive weeks in this epidemic period, and the alert week cannot be determined. ^5^ H1N1 and H3N2: influenza A (H1N1) and influenza A (H3N2) and B: influenza B.

**Table 4 ijerph-19-01677-t004:** Indicators to assess the model performance, stratified by the age groups.

Subtype	Age group	Se ^2^	Sp	PPV	NPV	YI	MT ^3^
Total ^4^	General ^1^	0.83	0.97	0.95	0.89	0.80	1
	Pre-school	0.82	0.97	0.95	0.90	0.79	1
	School-aged	0.88	0.97	0.96	0.91	0.85	1
	Adult	0.80	0.98	0.96	0.88	0.78	1
H1N1 ^5^	General	0.97	0.96	0.88	0.99	0.93	−1
	Pre-school	0.96	0.96	0.85	0.99	0.92	−1
	School-aged	1.00	0.96	0.88	1.00	0.96	0
	Adult	0.88	0.98	0.93	0.95	0.85	0
H3N2	General	0.97	0.97	0.97	0.97	0.94	0
	Pre-school	0.79	0.97	0.96	0.86	0.76	0
	School-aged	0.89	0.97	0.96	0.91	0.85	0
	Adult	0.84	0.95	0.95	0.85	0.79	1
B	General	0.70	0.98	0.93	0.86	0.67	1
	Pre-school	0.69	0.98	0.92	0.89	0.66	1
	School-aged	0.77	0.96	0.91	0.89	0.73	0.5
	Adult	0.65	0.98	0.92	0.88	0.63	1.5

^1^ General: general population. ^2^ Se: sensitivity. Sp: specificity. PPV: positive predictive value. NPV: negative predictive value. YI: Youden’s Index. ^3^ MT: median timeliness of the available seasons, including the training set and the test set; negative numbers represented the early detection weeks, while positive numbers represented the detection lag. ^4^ Total: total influenza strain. ^5^ H1N1 and H3N2: influenza A (H1N1) and influenza A (H3N2) and B: influenza B.

## Data Availability

The data presented in this study are available on request from authors affiliated with Hubei Provincial Center for Disease Control and Prevention. The data are not publicly available due to these data were archived by the CDC for internal use.

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
