# Peer review of "Applying the Moving Epidemic Method to Establish the Influenza Epidemic Thresholds and Intensity Levels for Age-Specific Groups in Hubei Province, China"

_ijerph, 2022, doi:10.3390/ijerph19031677_

Round 1
Reviewer 1 Report
The authors present a very interesting article about the use of MEM to find surveillance indicators for influenza virus detection rate in three age groups in the province of Hubei.
The high amount of test performed in the studied seasons allows the analysis with high accuracy and good validity.
I have son minor questions and suggestions to the authors:
Page 1 line 33: The sentence is a little bit confusing to me. Please, rewrite.
Page 1 line 40: What study do the authors refer? Ref 5? I suggest the authors to put the reference in the previous sentence and to rewrite the whole sentence.
Page 2 line 127: I suggest the author to include the type of mean, arithmetic or geometric (supposed arithmetic)…. 95% one side confidence interval of the arithmetic mean of the 30…..
Page 5 line 185. I do not understand what the authors want to say with ‘stable epidemic’
Page 10 table 4. MT means ‘Mean of timeliness’? Please review the footnote (3)
Discussion
You say that climate and economic differences among provinces could explain the different thresholds among Hubei and Guangdong. In large countries/regions with two waves, it is relative frequent that the two waves appear in two well defined territories. Do you know if the winter and summer waves in the four seasons with two waves were detected in most of the Hubei cities or just in some of them?. Please discuss it a little bit.
Author Response
Point 1: Page 1 line 33: The sentence is a little bit confusing to me. Please, rewrite.
Response 1: The sentence was rewritten. We clarified the “significant loss” as the losses on the economy and people’s health. The “influenza A(H1N1)pdm09” was changed to “the influenza A (H1N1) pandemic in 2009”. (Page 1 line 37)
Point 2: Page 1 line 40: What study do the authors refer? Ref 5? I suggest the authors to put the reference in the previous sentence and to rewrite the whole sentence.
Response 2: We found that the sentence was too long to only express the view that vaccination timing is important. We combined Ref 5 (early vaccination reduced the vaccination protective effect) and Ref 6 (untimely vaccination resulted in insufficient vaccination coverage ) into the sentence “However, accurate prediction of the start of the epidemic was correlated with the protective effect of the vaccine.” at Page 2 line 47.
Point 3: Page 2 line 127: I suggest the author to include the type of mean, arithmetic or geometric (supposed arithmetic)…. 95% one side confidence interval of the arithmetic mean of the 30…..
Response 3: Following your suggestion, the arithmetic mean was added to the paragraph at Page 4 line 145.
Point 4: Page 5 line 185. I do not understand what the authors want to say with ‘stable epidemic’
Response 4: At Page 5 line 206, we changed the “stable epidemic” into the length and the start week of the epidemic period. “Stable” means these two indicators were less varied between seasons in H1N1 and summer peak of H3N2 than influenza B and winter peak of H3N2.
Point 5: Page 10 table 4. MT means ‘Mean of timeliness’? Please review the footnote (3)
Response 5: The footnote of Table 4 (Page 11 line 320) and Table A6 in Appendix C (Page 19 line 538) were corrected. MT means “Median timeliness”. It means the median timeliness of the available seasons, including the training set and the test set
Point 6: You say that climate and economic differences among provinces could explain the different thresholds among Hubei and Guangdong. In large countries/regions with two waves, it is relative frequent that the two waves appear in two well defined territories. Do you know if the winter and summer waves in the four seasons with two waves were detected in most of the Hubei cities or just in some of them?. Please discuss it a little bit.
Response 6: We checked the dataset of each city and found that all studied cities in Hubei Province had a two peaks pattern with four seasons. In the discussion, we added a paragraph at Page 12 line 357 to describe the epidemiology distinction between provinces in China and the epidemiology similarity between cities in Hubei Province. We add a reference that shows the differences in epidemiological patterns of Northern, Middle, and Southern China. We tried to use these phenomena to prove the epidemiology differences between cities may not be as great as the differences at the provincial level. As a result, to provide the evidence of establishing the epidemic thresholds at the provincial level to further support our study.

Reviewer 2 Report
The introduction is too long, too many details, the objectives were not detailed
the recommendations in the introduction
the main conclusion are to establishing the school-based virology surveillance system , but the autor focused on the vaccination program among school-aged children
Author Response
Point 1: The introduction is too long, too many details, the objectives were not detailed.
Response 1: We appreciated your suggestion. We merged the unnecessary details and deleted the recurring study purpose. Transferred the geographical and demographical information from the Introduction section to the Material and Method section. In the last paragraph of the introduction, we reclaimed the objective and the main approaches of our study.
Point 2: the main conclusion are to establishing the school-based virology surveillance system , but the autor focused on the vaccination program among school-aged children
Response 2: We rewrite the conclusion to focus on establishing the thresholds among children, and emphasizing the importance of setting up the surveillance system targeting the school-aged children.

Reviewer 3 Report
Applying the Moving Epidemic Method (MEM) to establish the influenza epidemic thresholds and intensity levels for age-specific groups in Hubei Province, China manuscript is an interesting study. considering that other groups have performed the MEM to perform this statistical study in other regions can bring a new method to predict the peak in seasonal influenza. Facing the COVID-19 related pandemic, these types of studies can deem a light on other types of epidemic diseases in the future.
I have some minor suggestions
for the abstract:
"Cross- 22 validation process showed that the school-aged children has better performance among all age groups in most cases." this sentence is ambiguous. please define better and most cases. the abstract should be understandable even before reading the full manuscript.
there are some grammatical errors also present in the text such as in the mentioned sentence: "...school-aged children have better performance."
I suggest not using abbreviations in the abstract:" The epidemic thresholds for H1N1, H3N2 and B were determined" the "B" can be written as influenza B.
In the introduction also there is an abbreviation without explaining it. "pdm09" is not necessary as you have not used this in the later texts. you can just type "2009 influenza A (H1N1) pandemic"
It would be interesting if you could provide a similar plot like Figure 4 for the general population of Hubei Province for comparison purposes. you can plot them both (the school-aged children and total population) on the same graph.
Author Response
Point 1: "Cross- 22 validation process showed that the school-aged children has better performance among all age groups in most cases." this sentence is ambiguous. please define better and most cases. the abstract should be understandable even before reading the full manuscript.
Response 1: We appreciated your suggestion. We substitute the word “performance” with the “sensitivity”. The phrase “most cases” was further clarified in line 25. The sensitivity of the model established by school-aged children was higher than other age groups in total influenza, H1N1 and influenza B, while it was only lower than the general population group in H3N2.
Point 2: There are some grammatical errors also present in the text such as in the mentioned sentence: "...school-aged children have better performance."
Response 2: We reexamed the grammatical errors in the passage.
Point 3: I suggest not using abbreviations in the abstract:" The epidemic thresholds for H1N1, H3N2 and B were determined" the "B" can be written as influenza B. In the introduction also there is an abbreviation without explaining it. "pdm09" is not necessary as you have not used this in the later texts. you can just type "2009 influenza A (H1N1) pandemic"
Response 3: Thank you for your kindly suggestions, we checked and corrected all abbreviations which were not appropriate.
Point 4: It would be interesting if you could provide a similar plot like Figure 4 for the general population of Hubei Province for comparison purposes. you can plot them both (the school-aged children and total population) on the same graph.
Response 4: We added one figure of the general population in Figure 4. which shows 1 week of false alert in season 2018/2019 (Page 10 line 284).
